# Automated Task-Informed Document Retrieval on the COVID-19 Open Research Dataset Using Topic Modeling

**Christine Herlihy**[*] and **Yuelin Liu**[*]
Department of Computer Science
University of Maryland, College Park
College Park, MD 20740

## Abstract

The COVID-19 Open Research Dataset (CORD-19), a fast growing collection of biomedical research literature, was made available in March 2020 as an effort to facilitate research addressing the COVID-19 pandemic, along with 17 tasks specifying the research questions of interest. We propose an automated, task-informed document retrieval framework for CORD-19 that leverages latent factors learned through topic models to select a set of research articles most relevant to the tasks at hand. Compared to naïve keyword-based approaches, our approach broadens the qualification of relevance from the presence of specific terms to the activity of latent topics. We show that our approach provides an overlapping yet notably different set of selections, as the latent factors account for meaningful document-wise co-occurrences that individual keywords fail to capture. Upon both qualitative and quantitative examination of retrieval results, we further provide recommendations regarding task creation intended for unsupervised document retrieval in large, heterogeneous, natural language datasets.

## 1 Introduction

The collective efforts of clinical researchers and policymakers around the world to identify the root cause(s) and transmission mechanism(s) of the COVID-19 pandemic and develop effective intervention strategies to reduce morbidity and mortality, have resulted in an extremely large corpus of academic work in a relatively short time span.

The dynamic nature of this corpus, and the extreme sense of urgency guiding COVID-19-related research as the disease continues to spread in the absence of a vaccine, has the potential to disrupt the slower-paced, iterative, and prior-informed nature of the peer-review process. Additionally, many

of the clinical experts who would be best-equipped to participate in annotation-based approaches to information retrieval are on the frontlines of the pandemic. These unprecedented circumstances motivate the use of imperfect but scalable, automated approaches to latent topic discovery, and mapping of clinical research questions to relevant document response sets, as an interim solution which can be iteratively improved via expert curation and validation over time.

To this end, we develop a task-informed document retrieval pipeline. Like examples in question answering systems, our approach follows the reasonable assumption that documents that best address a given task will share a similar set of latent concepts covered by the task description (Celikyilmaz et al., 2010). Leveraging non-negative matrix factorization (NMF), a common topic modeling technique, on CORD-19, we describe each research article or task description as a mixture of a set of latent topics. We then develop a relevance metric to map task documents to response documents based on the Jensen-Shannon divergence of each document's distribution over topics.

We demonstrate how this pipeline can be used to expand upon keyword-based approaches to corpus downselection and approximate task-specific document relevance in the absence of ground-truth labels. Such a task-informed document retrieval framework can serve as the basis for a structured, task-driven exploration of the CORD-19 corpus.

## 2 Data

### 2.1 The CORD-19 Corpus

The COVID-19 Open Research Dataset (CORD-19), launched in March 2020 by Allen Institute for AI and collaborators, is a fast-evolving set of academic literature concerning all aspects of COVID-19 and related topics (Wang et al., 2020).

---

Both authors contributed equally to this work.

Version 31 (updated June 18, 2020) of the CORD-19 corpus, $\mathcal{C}$, consists of more than $157,000$ articles about the novel coronavirus (SARS-CoV-2), the disease it causes (COVID-19), and related viruses, including but not limited to: SARS-CoV, which causes severe acute respiratory syndrome (SARS); MERS-CoV, which causes Middle East respiratory syndrom (MERS); and Ebola (Wang et al., 2020). We perform a series of domain-specific preprocessing and filtering steps, as detailed in Section 3.1.2, to prepare our experimental corpus $C \subsetneq \mathcal{C}$ for topic modeling.

## 2.2 The CORD-19 Tasks

The CORD-19 tasks are the set of 17 COVID-19-specific information retrieval and text summarization questions which the international scientific community has designated as both high-priority and amenable to discovery via text mining of the CORD-19 corpus (Wang et al., 2020). Each task consists of a high-level objective, and several specific sub-tasks. Research areas represented in the set of high-level objectives include: (1) the epidemiological dynamics of COVID-19; (2) patient- and population-level risk factors; (3) vaccines and therapeutics; (4) public health and public policy; and (5) information dissemination and scientific collaboration.

## 3 Methodological Approach

### 3.1 Preprocessing

#### 3.1.1 Keyword Selection

While comprehensive in its coverage of historical and COVID-specific epidemiological topics, CORD-19 contains entries that are less likely to address the COVID-focused tasks in question than others. To reduce the computational resources needed for our experiment, we identify two sets of keywords of interests to serve in a heuristic filtering of the entries in the original dataset. The containment of any of the keywords in its title is one of the inclusion criteria for a given document, and we will elaborate on the filtering steps in Section 3.1.2

The first set of keywords are selected to ensure that we include documents explicitly addressing our entities of interest. The set used in our experiment is enumerated in Table A.1.

The second set of keywords are designed to be both task-informed and domain-specific, ensuring that we include documents that may contain strong signals relevant to the tasks at hand, regardless of whether they explicitly address our entities of interest. To identify this set of keywords, we perform named entity recognition using scispaCy's en_core_sci_sm language model on each task description (Neumann et al., 2019). After a tokenization step that preserves the recognized entities, we perform term frequency zero-corrected inverse-document frequency (TF-IDF) transformation of the token count matrix. For each task, we identify 10 terms with the highest TF-IDF adjusted weights. The union of the top terms for each task (see Table A.2) make up our second set of keywords.

#### 3.1.2 Filtering

To prepare the CORD-19 corpus for topic modeling, we perform several rounds of filtering and text cleaning. We choose to use the title and abstract of a research article as a proxy for its content; as such, we filter the entries in CORD-19 to select documents containing both a title and an abstract. Since there are a notable amount of duplicate documents within the dataset, we excluded redundant entries based on their cord_uid, title, and abstract content. Next, we filter by the presence of any keyword identified in Section 3.1.1, after which we heuristically prune out non-English entries using naïve regular expression matching on common words in English. The resulting experimental corpus, $C$ has $59,635$ documents.

#### 3.1.3 Text Cleaning

To form the proxy document for each entry that remain after our filtering steps, we remove the section headers - if present - from its abstract, then concatenate its title and the abstract. For text cleaning, we first replace the hyphens in the text with underscores to preserve hyphenated entities, such like covid-19. Then, we remove non-alphanumerics, stopwords, and pure numerics from the text before turning it into lowercase. We perform the same text cleaning steps on the task descriptions.

To identify meaningful collocations, we train a gensim phraser using the normalized pointwise mutual information (NPMI) scoring metric on the combination of CORD-19 title-abstract document corpus and the task description corpus (Bouma, 2009). The bag-of-words representations for the CORD-19 entries and task descriptions are generated from the respective phrased texts for downstream transformations in preparation for topic modeling.

## 3.2 Task-Informed Document Retrieval

Our task-informed document retrieval pipeline pipeline takes the bag-of-words representations of the tokenized, preprocessed document and task corpora as input, and computes the TF-IDF transformation of the document corpus [1]. We then use NMF to decompose the TF-IDF transformed word-document matrix, $\mathcal{V}_\mathcal{C}$, associated with the document corpus into two low-rank, non-negative approximation matrices, $\mathcal{W}_\mathcal{C}$ and $\mathcal{H}_\mathcal{C}$, which represent the word-topic matrix and topic-document matrix, respectively (Zhao and Tan, 2017).

Next, we project the TF-IDF transformed word-document matrix, $\mathcal{V}_\mathcal{T}$, associated with the task corpus into the lower-dimensional space represented by the word-topic matrix $\mathcal{W}_\mathcal{C}$, and iteratively update the model to produce $\mathcal{H}_\mathcal{T}$, a topic-task matrix which represents each task as a distribution over topics. To retrieve the top-$n$ most relevant documents for each task $t$, we compute the pairwise Jensen-Shannon divergence between the task's topic distribution and the topic distribution of each document $c_i \in C$, sort ascending, and return the $n$ least divergent documents, which constitute a relevant response set for task $t$. Finally, the disjoint union of response sets over all tasks is returned. This pipeline is formalized in Algorithm 1 below:

---

**Algorithm 1** Task-Informed Document Retrieval

    **procedure** GETRELDOCS($\mathcal{C}, \mathcal{T}, k, n$)

        $\mathcal{V}_\mathcal{C} \leftarrow \text{TF-IDF} \circ \varphi(\mathcal{C})$

        $\mathcal{V}_\mathcal{T} \leftarrow \text{TF-IDF} \circ \varphi(\mathcal{T})$

        $W_\mathcal{C}, H_\mathcal{C} \leftarrow \text{NMF}(\mathcal{V}_\mathcal{C}, k)$

        $H_\mathcal{T} \leftarrow \psi(\mathcal{V}_\mathcal{T}, W_\mathcal{C}, H_\mathcal{C})$

        **for** $t \in \mathcal{T}$ **do**

            $D \leftarrow \{\text{JSD}(h_{\mathcal{T}*,t} \parallel h_{\mathcal{C}*,i})_i \mid i \in \mathbb{N}^{<|\mathcal{C}|}\}$

            $D \leftarrow \text{SORT}(D, \texttt{ascending})$

            $I \leftarrow \{i \mid d_i \in D[\texttt{:n}]\}$

            $D_{r_t} \leftarrow \{c_i \mid i \in I \wedge c_i \in C\}$

        **return** $\bigsqcup_{t=1}^{|\mathcal{T}|} D_{r_t}$

---

The parameters and subroutines are defined as follows: (1) $\mathcal{C}$ represents the CORD-19 document corpus; (2) $\mathcal{T}$ represents the CORD-19 task corpus; (3) $k$ represents the number of topics; (4)

$n$ represents the number of relevant documents to retrieve for each task; (5) $\varphi$ represents the series of preprocessing and filtering operations outlined in Section 3.1; (6) $\psi$ represents projection of $\mathcal{V}_\mathcal{T}$ to the lower-dimensional word-topic space; and (7) JSD is the pairwise Jensen-Shannon divergence (Virtanen et al., 2020). $\mathcal{V}_\mathcal{C} \in \mathbb{R}_{\geq 0}^{|V| \times |C|}$ and $\mathcal{V}_\mathcal{T} \in \mathbb{R}_{\geq 0}^{|V| \times |T|}$ represent the TF-IDF transformed word-document matrices constructed from the preprocessed document corpus and task corpus, respectively. $W_\mathcal{C} \in \mathbb{R}_{\geq 0}^{|V| \times k}$ is the word-topic matrix. $H_\mathcal{C} \in \mathbb{R}_{\geq 0}^{k \times |C|}$ and $H_\mathcal{T} \in \mathbb{R}_{\geq 0}^{k \times |T|}$ are the topic-document matrices associated with the document and task corpora. $D_{r_t}$ is the document response set associated with task $t$; $|D_{r_t}| = n$.

## 4 Empirical Evaluation

### 4.1 Hyperparameter Tuning

To evaluate our task-informed document retrieval pipeline on the CORD-19 document and task corpora, we design a set of experiments intended to help us select the topic model hyperparameter $k$ (the number of topics) such that the resulting NMF topic model yields semantically coherent topics.

### 4.1.1 Topic Model Coherence

We use Röder et al.'s $C_V$ metric, which has been empirically confirmed to be positively correlated with human annotators' judgements about the semantic coherence of topics, to evaluate the coherence of the topics produced by the NMF model trained on the CORD-19 corpus, and select a $k$ for which coherence is maximized (Röder et al., 2015).

Per Röder et al., the $C_V$ metric for a single topic, $t$, associated with a set of top words, $W$, where $|W| = n$, is computed in a four-step process: first, the set of top words, $W$, is segmented into top word subsets to produce a set, $S$, of pairs with left projection := a single word, $w_i \in W$, and right projection := the set of top words, defined as:

$$S = \{(W', W^*) \mid W' = \{w_i\}; w_i \in W; W^* = W\} \quad (1)$$

Next, a set of Boolean documents is created by sliding a window of length 110 over the tokenized reference corpus, advancing one token at a time. These pseudo-documents are used to estimate the occurrence and co-occurrence probabilities of word(s) in the set of top words, $W$.

Then, for each pair $S_i = (W', W^*) \in S$, a context vector is computed for $W'$ by mapping $w_i$

---

[1] https://radimrehurek.com/gensim/models/tfidfmodel.html

to each word $w_j \in W$ and computing the pairwise NPMI for each of the resulting pairs $(w_i, w_j)$. A context vector is similarly computed for $W^*$, with the modification that each entry is a sum of the pairwise NPMI calculations for each word $w_k \in W^*$ and $w_j \in W$. The cosine similarity of the resulting context vectors is then computed, and can be interpreted as a measure of the extent to which the words $\in W^*$ support word $w_i \in W'$ (Röder et al., 2015; Syed and Spruit, 2017).

The arithmetic mean of the cosine similarity scores associated with each element $S_i \in S$ is then computed to yield $C_{V_t}$, the $C_V$ coherence score for topic $t$. Finally, an overall coherence metric for the model, $C_V$, is computed by taking the arithmetic mean over topics: $\frac{1}{k}\sum_{t=1}^{k} C_{V_t}$.

The recency of COVID-19 and the domain-specific nature of the CORD-19 corpus complicate the task of identifying a semantically similar external reference corpus. As such, we treat $C_V$ as an intrinsic evaluation metric and use the CORD-19 corpus as the reference corpus from which to create Boolean sliding window documents. We compute NMF model coherence scores with $n = 10$ and values of $k$ between $[15, 125]$ with a step size of 5; the results are displayed in Figure 1.

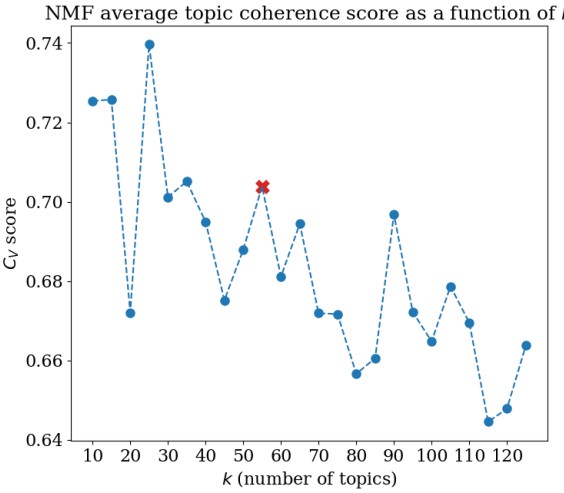

Figure 1: NMF $C_V$ score as a function of $k$

Our task-informed document retrieval end goal motivates the choice of a $k$ which strikes a balance between high coherence and topic granularity, since many of the tasks are relatively specific and are thus unlikely to be addressable with response sets drawn from overly generic topics. We thus treat $k \leq 35$ as a burn-in period, and select the $k$

associated with the maximum $C_V$ score over the interval $[40, 125]$, which is 55. The 10 highest-weighted terms associated with each of the model's 55 topics appear in Supplemental Table A.3.

## 4.2 Task-Document Response Set Results

### 4.2.1 Keyword-Based Approach as Baseline

We compute task-document response sets using a naïve approach based on keyword matching as a baseline to which we compare the results of our topic-modeling-based approach.

To obtain a more fine grained set of task-specific keywords for benchmarking, we apply a gensim phraser (as described in Section 3.1.3) and a gensim TF-IDF model [2] trained on the CORD-19 corpus to the task description documents preprocessed using steps mentioned in Section 3.1.3, producing a list of tokens ranked by their TF-IDF weights for each task. We also generate a list of task-specific stopwords using a weighting scheme which accounts for term frequency, document frequency, and inverse token length [3]. For each task, we select the top tokens ranked by the TF-IDF model that are not in the task-specific stopword list, and the number of tokens chosen is proportional to the task document length [4]. We also artificially inject "COVID-19" and "SARS-CoV-2" into each task-specific keyword set.

For each task-specific keyword set, we compute the number of exact matches each entry in the CORD-19 title-abstract document corpus has with the set. The total keyword count in a document for a given task is then normalized with respect to the length of the document. The resulting normalized keyword count for each of the 17 CORD-19 tasks then serves as the ranking metric used in our keyword-based document retrieval baseline approach.

### 4.2.2 Task Topic Distribution

The topic weight distributions for 17 CORD-19 task description documents learned via procedure described in Section 3.2 are shown in Figure 2. While the topic weight vector for each individual task remains sparse, and most task topic distributions are pairwise dissimilar, there do exist clusters

---

[2] https://radimrehurek.com/gensim/models/tfidfmodel.html

[3] $\text{weight}_w = \frac{TF_w \times DF_w^\alpha}{len(w)^\beta}$, where $\alpha$ and $\beta$ are parameters. For our experiment, we have $\alpha = \beta = 5$.

[4] In our experiment, we choose a keyword set for a task document of size $\frac{1}{5}$ of the number of tokens in the document.

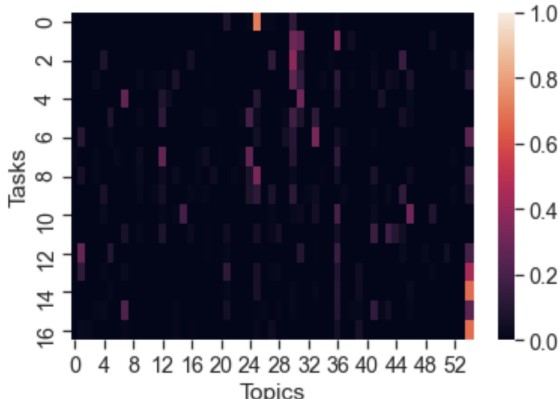

Figure 2: Topic weight distribution for 17 CORD-19 tasks across 55 topics learned by NMF.

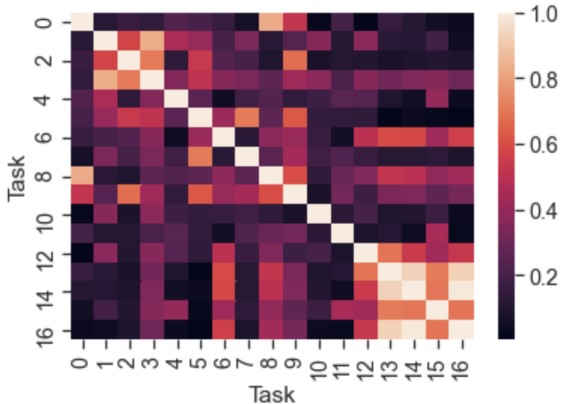

Figure 3: Pairwise cosine similarity between topic weight distributions learned by NMF for 17 CORD-17 tasks.

of topics (such as Task 14 and Task 16) for which their pairwise cosine similarity is $> 0.8$ (Figure 3).

Upon further observation on Task 14 and 16 by their top 10 domain-specific keywords (Table A.2) and top words from their most active topic (Figure 2, Table A.3), we see that both tasks seek to address public health, social, information, and psychological concerns of the pandemic. This witnesses the property of the CORD-19 tasks that they are not made to be pairwise disjoint thematically.

### 4.2.3 Task-Informed Retrieval Results

Our topic-modeling-based approach produces document response sets for each task $\in \mathcal{T}$ in accordance with the procedure described in Section 3.2. Within a response set, documents are represented as their distributions over topics, and ranked ascending in accordance with their Jensen-Shannon divergence with respect to the associated task document's distribution over topics.

The intuition behind this approach is that many

of the COVID-specific research questions posed by clinical researchers and policymakers are interdisciplinary in nature, and/or may benefit from the transfer learning potential contained within documents about historical pandemics and similar viral diseases. As such, this approach can be viewed as an effort to expand upon the more restrictive keyword-based approach, potentially trading off precision for recall in an effort to facilitate knowledge discovery vis-à-vis the jointly learned latent topics.

While we are not able to quantitatively compare the keyword and topic-modeling based approaches on the basis of precision and recall in the absence of relevance labels, we do inspect the title, normalized keyword count, and Jensen-Shannon divergence for the top 300 documents in the union and intersection of the response sets returned by each approach for each task.

Figure 4 visually represents the normalized keyword counts of each task's top 300 documents returned solely by the keyword-based approach, solely by the topic-modeling approach, and jointly by both approaches, while Figure 5 visually represents the Jensen-Shannon divergence associated with the response documents produced by each method, as well as documents in the intersection.

By virtue of the ranking metric associated with each approach, the documents chosen solely by the keyword-based approach have higher normalized keyword counts, and those chosen solely by Jensen-Shannon have lower Jensen-Shannon divergence scores than their keyword-based counterparts.

We do, however, observe that the intersection of the two response sets is non-empty for most tasks, and that the documents in the intersection tend to have higher keyword counts (e.g., they resemble documents selected solely via keywords), and lower Jensen-Shannon divergence scores (e.g., they resemble documents selected via topic modeling). We also note that: (1) for any specified response set size, the size of the intersection varies by task; and (2) the size of the intersection increases as the number of top-ranked documents returned is incremented.

With respect to the titles of the top documents, we have included a subset of the results for Task 0, which focuses on social distancing, contact tracing, and transmissibilty, and where both methods successfully return relevant documents (see Table 1), and Task 1, which focuses on the hypercoagulable

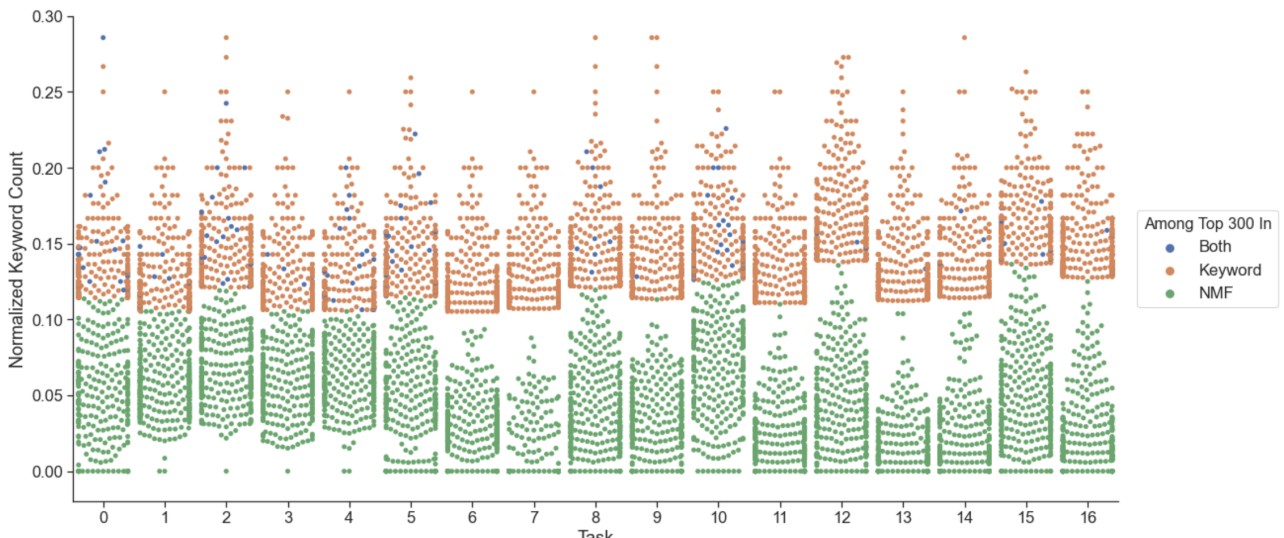

Figure 4: Count of task-specific keywords in a document, normalized by document length, for top 300 documents retrieved by normalized keyword count (orange) and/or NMF topic-based divergence metric (green). Documents in blue are among the top 300 in both lists.

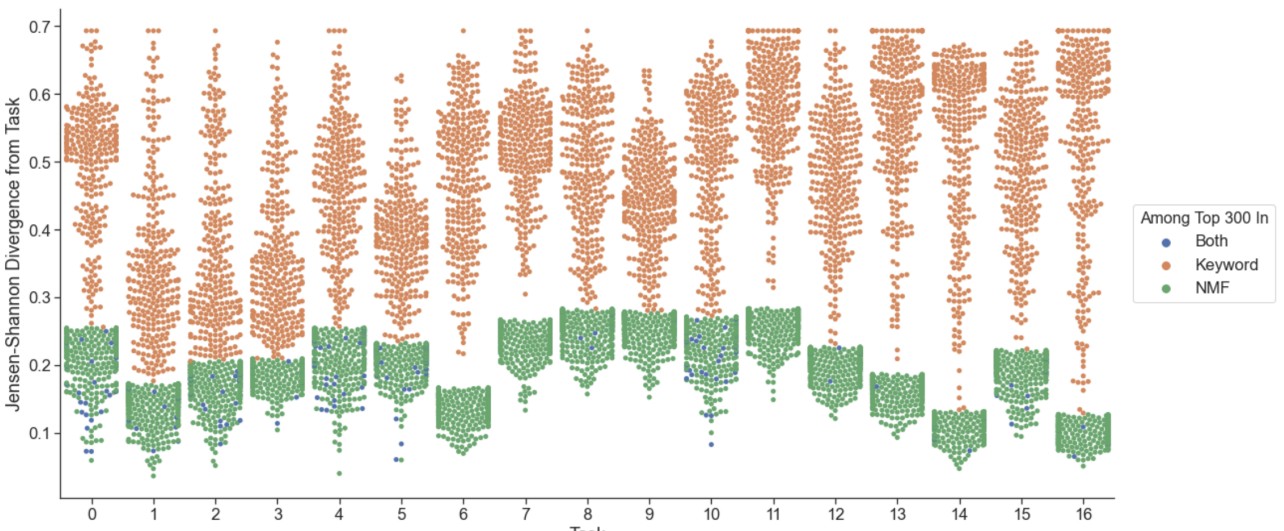

Figure 5: Jensen-Shannon Divergence between document topic weight vector and task topic weight vector learned by NMF, for top 300 documents retrieved by normalized keyword count (orange) and/or NMF topic-based divergence metric (green). Documents in blue are among the top 300 in both lists.

| Method | Top 3 Titles Returned on CORD-19 Task 0 |
|---|---|
| Keyword | COVID-19: Transmission, prevention, and potential therapeutic opportunities |
| | Application of personal-oriented digital technology in preventing transmission of COVID-19, China |
| | COVID-19 Trend and Forecast in India: A Joinpoint Regression Analysis |
| Topic Modeling | Contact Tracing: a game of big numbers in the time of COVID-19 |
| | Weathering COVID-19 Storm: Successful Control Measures of Five Asian Countries |
| | Quantifying interpersonal contact in the United States during the spread of COVID-19: first results from the Berkeley Interpersonal Contact Study |
| Both | Effectiveness of isolation, testing, contact tracing and physical distancing on reducing transmission of SARS-CoV-2 in different settings |
| | Assessing the plausibility of subcritical transmission of 2019-nCoV in the United States |
| | Contact tracing strategies for COVID-19 containment with attenuated physical distancing |

Table 1: Top 3 titles returned by keyword-based, topic-modeling-based methods, and both on CORD-19 Task 0.

state associated with COVID-19 and novel therapeutics (see Table 2). Neither method returns documents focused on hypercoagulation, but both methods return documents focused on COVID-related clinical trials and therapeutics. We hope to investigate the relationship between task characteristics

| Method | Top 3 Titles Returned on CORD-19 Task 1 |
|---|---|
| kw | Association of Use of Angiotensin-Converting Enzyme Inhibitors and Angiotensin II Receptor Blockers With Testing Positive for Coronavirus Disease 2019 (COVID-19). |
| | COVID-19 in India: Are Biological and Environmental Factors Helping to Stem the Incidence and Severity? |
| | Complement Activation in Patients with COVID-19: A Novel Therapeutic Target |
| nmf | Ethics of clinical trials |
| | Coronavirus disease 2019: a bibliometric analysis and review. |
| | Coronavirus disease (COVID-19): a scoping review |
| both | A systematic review of the prophylactic role of chloroquine and hydroxychloroquine in coronavirus disease-19 (COVID-19) |
| | Review of the Clinical Characteristics of Coronavirus Disease 2019 (COVID-19) |
| | Chest CT and Coronavirus Disease (COVID-19): A Critical Review of the Literature to Date. |

Table 2: Top 3 titles returned by keyword-based, topic-modeling-based methods, and both on CORD-19 Task 1.

and the precision and recall associated with the response sets returned by each of our retrieval approaches in future work.

## 5 Recommendations for Task Creation

Given that un- and weakly-supervised approaches to text mining the COVID-19 literature are likely to dominate research in this space over the near-to-medium term as uncertainty about the virus persists, we endeavor to offer recommendations about how future tasks can be structured from a lexical, syntactic, and semantic perspective to facilitate the success of automated, weakly supervised, task-informed retrieval methods like the one we have outlined here, in the hopes that they may be useful in future crowd-sourced research efforts.

We observe that our keyword-based baseline and our task-informed, topic-modeling based approach both struggle to return qualitatively relevant top-ranked documents for lexically sparse tasks with key terms that are rare within the CORD-19 corpus, and/or isolated within the task sentence(s), as opposed to co-occurring with synonyms.

One example is Task 1, which contains only two subtasks and a rare keyword (hypercoagulable): "(1a) What is the best method to combat the hypercoagulable state seen in COVID-19? (1b) What is the efficacy of novel therapeutics being tested currently?" The term "hypercoagulable" occurs infrequently in $C$, and subtask (1a) does not contain any supporting clinical synonyms (e.g., thrombosis, blood clot, etc.) that could help to bias the model toward more relevant documents.

Challenges also arise when lexically sparse tasks or subtasks contain words with ambiguous senses, particularly when a dis-preferred sense dominates within the corpus used to train the topic model. The use of the word "method" in subtask (1a), for example, is intended to refer to clinical methods, but the associated distribution over topics indicates that the presence of this term may have biased the model toward topics associated with quantitative

methods and models.

From a syntactical and semantic perspective, our document retrieval approaches qualitatively perform best when the subtasks within a task are grammatically self-contained and semantically homogeneous or complementary. By self-contained, we refer to subtasks which restate or propagate COVID-specific dependencies down from the "root" of the task, as opposed to subtasks with implicit dependencies that are not made explicit. Task 4 is an instructive example of the former: the root task asks researchers to, "Create summary tables that address diagnostics studies related to COVID-19", while two subtasks focus on: "(4a) Diagnosing SARS-CoV-2 with Nucleic-acid based tech; and (4b) Diagnosing SARS-CoV-2 with antibodies".

Semantically homogeneous tasks include include those with a unifying main question and hyponymic subtasks (e.g., subtasks which focus on or enumerate specific patient populations, diagnostic tests, and/or risk factors). Semantically complementary tasks include subtasks which are distributionally related, and may be expected to co-occur within the literature. Examples include Task 9, which has subtasks focused on both clinical and behavioral risk factors, as well as clinical and public-policy oriented risk mitigation strategies.

With these characteristics in mind, associated recommendations for future task creation include: (1) select semantically critical key terms which are unambiguous when possible; (2) consider augmenting via selective expansion of key terms, and/or ontological grounding of clinical terms; (3) make semantic dependencies and/or contextual information lexically and syntactically explicit in subtasks; and (4) break long, semantically heterogeneous tasks into their component pieces when possible.

## 6 Conclusions

In this work, we proposed a task-informed document retrieval framework for CORD-19 that leverages the learned latent topics from the corpus. Due

to the interpretability of its results, topic modeling enables automated quantitative and qualitative characterization of the documents in the corpus. We explored employing such characterizations in downstream document retrieval tasks, and compared the performance of such an approach against a naïve keyword-based approach.

We demonstrate that the topic modeling-based approach provides great potential in serving in unsupervised, automated document retrieval for a set of well-defined tasks, and provide recommendations regarding task creation for future crowdsourced, unsupervised retrieval efforts. In future work, we hope to improve our approach and provide quantitative analysis of the results via: (1) supervised learning, as corpora labelled with relevance measure(s) become available [5]; and (2) expert-in-the-loop keyword curation, informed prior integration, topic model tuning, and response set validation.

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

## A Appendices

| Keyword Set 1 |
| --- |
| covid19  covid-19  corona virus coronavirus  sars-cov-2  sars-cov sars  2019-ncov  mers-cov mers  wuhan virus  wuhan severe acute respiratory syndrome middle east respiratory syndrome |

Table A.1: One of two sets of keywords used in filtering the entries in the original CORD-19 dataset.

---

[5] https://ir.nist.gov/covidSubmit/data.html

| | Keyword Set 2 |
|---|---|
| Task 0 | effectiveness, workplace distancing, inter inner travel, strategy, significant, seasons, school, restriction, reduction, quarantine |
| Task 1 | tested, seen, combat, hypercoagulable state, method, currently, novel, efficacy, therapeutics, best |
| Task 2 | risk factor, related, covid 19, cerebrovascular disease, diabetes, cancer, cardio, respiratory system diseases, male, hypertension |
| Task 3 | mutations, covid 19, studies, virus, infector infectee pair, open questions, regional, adaptations, genetic variations, known |
| Task 4 | diagnosing, sars cov 2, disease presentations, advances, tech, antibodies, nucleic acid, based, new, point of care test |
| Task 5 | incubation, viral shedding, duration, asymptomatic, period, patients, virus, onset, length, proportion |
| Task 6 | patients, underhoused, protect, create, combating, socioeconomic status, lower, resource failures, outbreaks, hospital infrastructure |
| Task 7 | know, viral shedding, surfaces, cleaning agents, virus persist, inanimate surfaces, phobic, remain, nasopharynx, viable |
| Task 8 | transmission, health care, community settings, persistence, virus, effectiveness, disease, asymptomatic shedding, environment, movement |
| Task 9 | virus, transmission, including, fatality, behavioral factors, respiratory viral infections, transmissible, hospitalized, pregnant women, pre existing |
| Task 10 | vaccine, animal models, therapeutics, prioritize, discover, efforts, evaluate, approaches, include, studies |
| Task 11 | evidence, spill over, farmers, infected, receptor binding, nagoya protocol, sets, livestock, field surveillance, serve |
| Task 12 | outcomes, data, workforce, efforts, manifestations, practices, clinical, covid 19, people, best |
| Task 13 | npis, costs, research, policy, experiments, guidance, critical, resources, time, compliance |
| Task 14 | efforts, identify, measures, public health, ethics, existing, articulate, thematic areas, fuel, engage |
| Task 15 | testing, future, including, understanding, diagnostics, use, policies, pathogens, technology, detect |
| Task 16 | disadvantaged, public, populations, capacity, information, gaps, communication, non federal, federal state local public health surveillance systems, citizens |

Table A.2: Top 10 tf-idf keywords identified in Kaggle tasks after Scispacy en_core_sci_sm language model.

| Topic | Top 10 Terms |
| --- | --- |
| 0 | h1n1, pandemic, influenza, virus, pdm09, swine, mexico, oseltamivir, seasonal, infection |
| 1 | care, icu, intensive, critical, unit, cmv, patient, palliative, workers, critically_ill |
| 2 | mhv, rna, structure, protein, domain, information, synthesis, structural, replication, sequence |
| 3 | ebola, ebov, evd, west, calves, africa, response, immune, protection, outbreak |
| 4 | lung, transplantation, transplant, stem, liver, transplant_recipients, recipients, disease, organ, hematopoietic |
| 5 | patients, injury, ecmo, aki, failure, kidney, oxygenation, extracorporeal, renal, patient |
| 6 | pulmonary, lung, pressure, hypertension, hiv_1, blood, mucosal, vaccine, chest, responses |
| 7 | detection, assay, rt_pcr, testing, diagnostic, sensitivity, assays, real_time, diagnosis, rapid |
| 8 | igg, elisa, antibody, samples, igm, serum, antibodies, sera, assay, sars |
| 9 | mice, mouse, mhv, cells, demyelination, cd8, cns, hepatitis, virus, infection |
| 10 | hiv, aids, immunodeficiency, prrsv, hiv_1, people, antiretroviral, africa, media, hiv_infected |
| 11 | h7n9, prrsv, vaccine, vaccines, virus, dna, reproductive, avian, human, influenza |
| 12 | entry, fusion, viral, proteins, replication, viruses, virus, sars_cov, membrane, host |
| 13 | antibodies, cells, monoclonal, antibody, human, cell, ace2, tgev, receptor, binding |
| 14 | immune, cells, innate, response, ifn, replication, interferon, responses, activation, expression |
| 15 | vaccine, dengue, vaccination, vaccines, development, denv, responses, immune, vaccinated, vector |
| 16 | ace2, cardiovascular, pregnancy, hypertension, receptor, angiotensin_converting_enzyme, pregnant_women, angiotensin, pneumonia, ace |
| 17 | zikv, zika, dengue, microcephaly, pregnancy, pregnant_women, virus, infection, fetal, neurological |
| 18 | rna, rnas, viruses, virus, viral, pcr, gene, method, detection, assay |
| 19 | children, pediatric, adults, hospitalized, pneumonia, infants, age, years, clinical, hbov |
| 20 | hcv, hepatitis, liver, virus, replication, 229e, chronic, cd81, ns5a, anti_hcv |
| 21 | students, cases, school, surveillance, data, closure, case, schools, training, symptoms |
| 22 | hbv, expression, genes, gene, ace2, data, hepatitis, analysis, tmprss2, mirnas |
| 23 | cats, feline, fip, fcov, peritonitis, fipv, felv, fiv, cat, dogs |
| 24 | samples, viruses, respiratory, isolates, detected, viral, pcr, pneumoniae, asymptomatic, specimens |
| 25 | transmission, contact, sars_cov_2, tracing, contacts, quarantine, measures, social_distancing, strategies, isolation |
| 26 | influenza, viruses, surveillance, season, ili, seasonal, virus, h3n2, seasons, influenza_like |
| 27 | stroke, ischemic, risk, endovascular, thrombectomy, score, outcome, acute, occlusion, patients |
| 28 | model, epidemic, number, models, dynamics, parameters, data, reproduction, mathematical, peak |
| 29 | group, groups, laparoscopic, patients, postoperative, pain, mesh, significantly, hernia, repair |
| 30 | covid_19, china, cases, outbreak, wuhan, pandemic, transmission, spread, healthcare, measures |
| 31 | sars_cov_2, covid_19, severe, coronavirus, patients, pneumonia, symptoms, syndrome, chest, clinical |
| 32 | rsv, syncytial, infants, respiratory, hmpv, bronchiolitis, virus, metapneumovirus, hospitalized, viruses |
| 33 | patients, cancer, treatment, management, pneumonia, diagnosis, guidelines, recommendations, covid_19, clinical |
| 34 | h5n1, avian, viruses, poultry, pathogenic, birds, influenza, human, virus, highly |
| 35 | cancer, lung, breast, tumor, prostate, expression, gastric, cancers, tmprss2, cells |
| 36 | review, studies, systematic, research, trials, evidence, literature, articles, meta_analysis, clinical |
| 37 | sars, respiratory, infections, tract, acute, syndrome, severe, sars_cov, infection, coronavirus |
| 38 | dna, sequences, sequencing, sequence, genome, denv, method, genomes, weight, samples |
| 39 | mers_cov, middle_east, health, public, countries, mers, global, international, syndrome, saudi_arabia |
| 40 | surgery, laparoscopic, surgical, resection, ablation, technique, complications, rectal, procedure, aneurysms |
| 41 | bats, diseases, bat, species, human, animal, infectious, emerging, zoonotic, humans |
| 42 | epitopes, mabs, neutralizing, antibodies, epitope, rbd, peptides, antigenic, monoclonal, sars_cov_2 |
| 43 | host, cell, autophagy, cells, apoptosis, interactions, pathogens, receptors, cellular, infection |
| 44 | tgev, strains, strain, genome, sequence, porcine, gene, virus, pigs, china |
| 45 | mortality, age, patients, risk, outcomes, icu, admission, hospital, years, associated |
| 46 | vaccines, network, models, networks, bovine, animal, calves, vaccine, development, cattle |
| 47 | pedv, diarrhea, porcine, epidemic, piglets, cells, pigs, ped, swine, vero |
| 48 | niv, ventilation, failure, intubation, respiratory, sepsis, oxygen, noninvasive, nipah, non_invasive |
| 49 | drugs, antiviral, drug, compounds, inhibitors, protease, activity, treatment, chloroquine, hydroxychloroquine |
| 50 | protein, sars_cov, spike, sars, proteins, nucleocapsid, recombinant, fusion, membrane, domain |
| 51 | ards, ecmo, distress, syndrome, survival, extracorporeal, respiratory, mortality, acute, oxygenation |
| 52 | asthma, exacerbations, copd, vaccination, exacerbation, chronic, influenza, obstructive, airway, infections |
| 53 | ibv, bronchitis, diabetes, infectious, chickens, strains, avian, chicken, strain, isolates |
| 54 | health, public, pandemic, mental, covid_19, social, crisis, care, services, psychological |

Table A.3: NMF Topic Model Top 10 Terms Per Topic ($k = 55$)

