# OpenReview forum: "Automated Task-Informed Document Retrieval on the COVID-19 Open Research Dataset Using Topic Modeling"
_EMNLP/2020/Workshop/NLP-COVID — Submitted to NLP-COVID19-EMNLP_

### Official Review · AnonReviewer2 · 2020-09-21
**Interesting work but not solid enough**

**Rating:** 5
**Confidence:** 4

**Review:**

The study proposed to use NMF to extract hidden topics from the COVID-19 Open Research Dataset  for 17 tasks, which was used for information retrieval for each task.  The method is quite standard and straight-forward.

Strength of this paper is : i)it showed the effectiveness of NMF over keyword approach.  ii) the examples given in section 5 are inspiring.

However, there are some weaknesses: i) The keyword approach is rather a simple baseline, authors may consider other vector space models or language models for IR. ii) For evaluation, the paper did some effort in assessing topics, it would be better to evaluate the IR results further. e.g., do some extrinsic evaluation on down stream tasks and see if the retrieved results are really beneficial; iii) Not sure whether the two key word sets are good or not. Especially in the second set, it returned some noisy words (e.g. best) using tfidf for key word selection.

Some questions for authors: In algorithm 1, what is ψ exactly ? how did you do the dimension projection (reduction) ?

---

> ### Author Response · Authors · 2020-09-28
> **Response to Reviewer2**
>
> Thank you for your time and feedback.
>
> We agree that a comparison to a vector-space model would strengthen the analysis. We were not aware of labeled datasets at the time this analysis was conducted, but do plan to extend our analysis by evaluating this approach against TREC-COVID. We did employ NER to encourage named entity ngrams as keywords, and use a stopword list during keyword selection in an effort to reduce noise, but agree that further pruning would be helpful.
>
> To address your questions:
> 1. In Algorithm 1, $V_\mathcal{T}$ represents the TF-IDF transformed word-document matrix associated with the CORD-19 task documents. NMF factorizes the pre-processed, TF-IDF transformed word-document matrix associated with the input corpus into two lower-dimensional matrices, $W_{\mathcal{C}}$ and $H_{\mathcal{C}}$, which represent the word-topic matrix and topic-document matrices, respectively. Thus, $\psi$ represents computing the normalized dot product of $W_{\mathcal{C}}^T$ and $V_\mathcal{T}$ to produce $H_{\mathcal{T}}$, the topic-document matrix associated with CORD-19 tasks. We implement this model using gensim.

---

### Official Review · AnonReviewer3 · 2020-09-22
**An interesting and creative approach to document retrieval but lacking adequate evaluation.**

**Rating:** 5
**Confidence:** 3

**Review:**

The paper presents a task-informed document retrieval framework for search over the CORD-19 dataset. The authors suggest that latent variables found in topic distribution of each task and document can contribute to the higher quality retrieval, specifically, they score each document by the JSD between its topic distribution and that of the task at hand. The approach is well described and sufficiently detailed throughout the paper, but lacks better motivation – why the authors think this method is likely to outperform the keyword-based search? Some concrete examples could be helpful. Otherwise, the paper is clear, coherent, and very well-written.

My main concerns about this work lie in two main aspects: (1) the lack of comparison to traditional methods in IR, e.g., vector space, probabilistic and language models, and (2) insufficient evaluation.

Re (1): Why using a somewhat simplistic keyword match model as the baseline while there exist established and successful approaches to IR? I found the proposed methodology creative and potentially novel and would be curios about how it compares to other IR techniques.

Re (2): Considering the two approaches presented in the paper, the evaluation (presented mainly in section 4.2.3) makes is difficult to draw any conclusions: figures 4-5 present highly expected results (“by the virtue of the ranking”); surprisingly little documents are in the overlap of the two approaches, which highlights the need in adequate evaluation; no actual comparison between the two approaches in conducted. The authors could use, e.g., the TREC-COVID labeled dataset for evaluation.

I thought that was an interesting and creative approach, and the paper was a very enjoyable read overall. The two issues above, however, prevent me from recommending it for acceptance. With an adequate evaluation (and proven benefits of the suggested approach) this could be a very good work.

Minor:
Section 4.1.1 could be shortened or omitted; topic model coherence seems a bit off-topic.

Figure 2 could be clearer if presented in negative (i.e., light) colors.

Page 3: “pipeline pipeline” -> “pipeline”

Page 7: “include include” -> “include”

Page 4: why 35 was chosen for k?

---

> ### Author Response · Authors · 2020-09-28
> **Response to Reviewer3**
>
> Thank you for your time and feedback.
>
> To address your questions:
> 1. We hypothesized that the JSD-based approach might outperform keyword-based search for CORD-19 task-specific document retrieval because of the heterogeneity of the CORD-19 corpus, and the complex, inter-disciplinary nature of many of the CORD-19 tasks. While a set of keywords can of course be used alongside conditional logic to represent this type of complexity, we felt that the ideal response set for many tasks would contain documents that were themselves multi-disciplinary in nature (e.g., drawn from a similar distribution over topics as the task in question), rather than isolated and specific, as reviewing documents that synthesized prior work and COVID-specific recommendations would be more efficient for scientific/clinical end-users.
> 2. We used a keyword match model as the baseline because we did not have access to a labeled dataset at the time this analysis was performed, and as such, were not able to employ supervised IR techniques or compute canonical evaluation metrics, such as precision, recall, or f-score.
> 3. We agree with your suggestion regarding Figure 2, and plan to update the color scheme. We've also fixed the typos.
> 4. To address your question regarding $k$: we chose k to maximize average topic coherence but did require topics to sufficiently granular upon manual review. We observed that while values of $k <35$ did result in slightly higher average topic coherence scores, the topics produced tended to be too high-level to be useful for differentiation of tasks when represented as distributions over topics. This hyperparameter selection question motivated our discussion of topic model coherence in Section 4.1.1, though we are certainly amenable to shortening that section.

---

### Official Review · AnonReviewer1 · 2020-09-25
**Document Retrieval Using Topic Modeling**

**Rating:** 5
**Confidence:** 4

**Review:**

The authors describe the development of a task-informed document retrieval framework that leverages latent factors learned through topic models. The framework is applied to the CORD-19 corpus and compared to a naıve keyword-based document retrieval approach. The authors note that they are not able to quantitatively compare the keyword and topic-modeling based approaches on the basis of precision and recall in the absence of relevance labels.

The key intent of the CORD-19 dataset was to facilitate automation of knowledge synthesis for specific questions. Given the exponentially increasing number of studies, automation of knowledge synthesis would require a model to, as precisely as possible, identify evidence that is most relevant and contains information which can be analyzed to provide answers to those questions. While the idea of topic modeling is interesting, the authors do not provide any indication that such an approach, as described, would be able to support any relevant evidence synthesis. Therefore, it is impossible to comment on the model and draw conclusions on its usefulness or advantages when compared to other information/document retrieval methods.

---

> ### Author Response · Authors · 2020-09-28
> **Response to Reviewer1**
>
> Thank you for your time and feedback.
>
> The problem our work directly addresses is task-informed document retrieval. We recognize that the key intent of the CORD-19 dataset was to facilitate evidence synthesis from an increasing number of studies, and we believe that high-quality document retrieval is incredibly valuable in an evidence synthesis pipeline: it not only can serve as a down selection step to reduce the computation load required for more sophisticated downstream evidence synthesis, but also has immediate applications to current non-automated knowledge retrieval process by domain experts in their scientific research process.
>
> More importantly, we recognize that CORD-19 provides a problem intrinsically different from traditional knowledge synthesis/information retrieval problems. Along with the research studies, the dataset includes tasks elaborated by subtasks and other descriptions, providing a richer semantic context for each task. Our work studies how we can leverage such context information - when it is available - in a document retrieval task. Our preliminary work shows advantages to include such context via topic modeling by comparing to keyword-based document retrieval.
>
> To that end, we further provide recommendations for document/information retrieval task writing, exploring ways to address issues such as word sense disambiguation from information contained in tasks. We imagine such an approach has immediate applications in aiding scientific research by domain experts, as scientific research questions could be complex, nuanced, and impossible to be sufficiently summarized by a handful of keywords. However, due to the lack of a labeled dataset with such rich task descriptions, we were unable to effectively benchmark this claim.